# New Frontiers in *Acanthamoeba* Keratitis Diagnosis and Management

**DOI:** 10.3390/biology12121489

**Published:** 2023-12-05

**Authors:** Omar Shareef, Sana Shareef, Hajirah N. Saeed

**Affiliations:** 1School of Engineering and Applied Sciences, Harvard College, Cambridge, MA 02138, USA; omarshareef@college.harvard.edu; 2Department of Bioethics, Columbia University, New York, NY 10027, USA; 3Department of Ophthalmology, University of Illinois Chicago, Chicago, IL 60607, USA; 4Department of Ophthalmology, Harvard Medical School, Boston, MA 02115, USA

**Keywords:** *Acanthamoeba* keratitis, diagnosis, treatment, management, machine-learning, PCR, antibody, confocal microscopy, culture

## Abstract

**Simple Summary:**

This literature review examines recent breakthroughs in diagnosing and treating *Acanthamoeba* keratitis (AK), a rare yet severe corneal infection, often associated with contact lens use. Until recently, the diagnosis and management of AK had remained stagnant for decades, while the virulence and severity of AK has increased. AK can have a severe course resulting in significant vision loss, and its severity is often associated with delayed diagnosis and/or recalcitrant disease. Recent advances in diagnostic accuracy and targeted management strategies hold promise in changing the trajectory of AK prognosis. This review focuses on the literature published in the last 5 years, and assesses current evidence and gaps in knowledge. Key findings include newer and more targeted PCR techniques for diagnosis, some of which are ready for clinical implementation, while antibody-based and machine-learning diagnosis is still in its infancy but holds promise for the future. New treatment strategies including oral medication and single-drug topical therapy may improve drug access, efficacy, and offer options in recalcitrant disease. As AK severity increases, the novel diagnostic and treatment strategies described in this review may improve outcomes for patients.

**Abstract:**

*Acanthamoeba* Keratitis (AK) is a severe corneal infection caused by the *Acanthamoeba* species of protozoa, potentially leading to permanent vision loss. AK requires prompt diagnosis and treatment to mitigate vision impairment. Diagnosing AK is challenging due to overlapping symptoms with other corneal infections, and treatment is made complicated by the organism’s dual forms and increasing virulence, and delayed diagnosis. In this review, new approaches in AK diagnostics and treatment within the last 5 years are discussed. The English-language literature on PubMed was reviewed using the search terms “*Acanthamoeba* keratitis” and “diagnosis” or “treatment” and focused on studies published between 2018 and 2023. Two hundred sixty-five publications were initially identified, of which eighty-seven met inclusion and exclusion criteria. This review highlights the findings of these studies. Notably, advances in PCR-based diagnostics may be clinically implemented in the near future, while antibody-based and machine-learning approaches hold promise for the future. Single-drug topical therapy (0.08% PHMB) may improve drug access and efficacy, while oral medication (i.e., miltefosine) may offer a treatment option for patients with recalcitrant disease.

## 1. Introduction

*Acanthamoeba* keratitis (AK) is a serious eye infection caused by the *Acanthamoeba* species of protozoa [1]. Infectious keratitis can result in corneal scarring or perforation, and if left untreated can result in permanent vision loss [2,3]. Symptoms of AK include a foreign body sensation of the eye, pain, redness, blurred vision, photophobia, and excessive tearing [4,5].

*Acanthamoeba* spp. are commonly found in the environment, particularly in water sources such as tap water, swimming pools, hot tubs, and soil [6,7]. There are at least 24 known species of *Acanthamoeba*, with keratitis being the ocular manifestation of *Acanthamoeba* infection [8]. While they are usually harmless, they can cause infections if they come into contact with the eye, typically through the use of contaminated water and contact lenses [1,9,10]. While cases can occur in non-contact lens wearers, contact lens wear appears to be a major risk factor for AK, particularly soft contact or orthokeratology lenses [11,12,13].

Increased AK awareness and diagnosis has resulted in the disease being increasingly recognized on a global scale [14,15,16,17,18]. However, there does not appear to be a globally systematic method of AK diagnosis and treatment, resulting in widespread variations in management worldwide [19]. Furthermore, because of its global prevalence, notable variations in risk factors and microbial epidemiology exist across different regions [11,20]. As a result, differences in the pathogenesis of the disease can lead to regional differences in treatment efficacy. Additionally, certain antimicrobial agents and diagnostic modalities are not universally accessible [19].

Diagnosing AK can be challenging, as its symptoms can mimic those of other eye infections, particularly in its early stages [21,22]. Furthermore, diagnostic methods may vary depending on the expertise of the healthcare team, the available resources, the urgency of diagnosis, and the specific clinical scenario. Lacking a rapid, highly sensitive and accurate test for AK, diagnosis of the disease can be difficult [23].

Furthermore, the treatment of AK is also difficult. *Acanthamoeba* exists in two main forms during its life cycle: the trophozoite, and the cyst [24]. As a trophozoite, the organism is in its active stage and can directly invade the cornea and cause infection. The cyst form, while not actively infectious, is crucial for the transmission and persistence of *Acanthamoeba* in the environment [25]. Moreover, the cystic form is more difficult to target and eradicate due to a protective cell wall, metabolic inactivity, and lower sensitivity to antimicrobials. If conditions become more favorable, cysts can transform back into the trophozoite form and initiate infection [26]. Given the morphologic variations of *Acanthamoeba*, prolonged treatment courses, as well as the difficulties in targeting the cystic form of the organism, the management of AK can prove difficult [27].

The need for new AK treatments and diagnostics arises from having limited current options, the potential for severe damage, the evolving risk factors, and resistance challenges. Innovative approaches are crucial to effectively manage the infection, prevent lasting consequences, and provide alternatives to existing treatments. Additionally, tailored strategies are required to address diverse strains and resistance patterns. Improved interventions can alleviate the overuse of current therapies and enhance patient outcomes. The goal of this review is to examine new approaches in the diagnostics and treatment of AK and to disseminate current advances in the management of AK to improve clinical outcomes for patients with the disease.

## 2. Methods

We performed a review of all of the appropriate literature by executing a search of all the English-language literature on PubMed, searching for publications that matched the search terms “*Acanthamoeba* keratitis” and “diagnosis” or “treatment”. Since the current review seeks to evaluate new advances in AK diagnosis and treatment, restrictions were placed on the studies’ dates and only studies from 2018 to 2023 were included in the review. The search yielded book excerpts, clinical trials, randomized controlled trails, case reports, case series, systematic reviews, literature reviews, and correspondences. Inclusion criteria included publications within the timeframe above which presented new patient data or highlighted relevant information regarding AK diagnosis or treatment. Studies need not have been conducted in humans to be included in the review. Exclusion criteria included publications which solely reviewed cases or data outside of the study timeframe. The search criteria returned a total of 265 publications. Eighty-seven of the 265 publications met the inclusion and exclusion criteria and were included in this review. Other articles which were not directly found through the search criteria have been included in this review for additional discussion and insight into AK diagnosis and treatment.

## 3. Diagnosis

Diagnosing AK involves a combination of a clinical evaluation, a thorough review of patient history, and specialized laboratory and ancillary tests due to its challenging and often nonspecific symptoms. There exist several commonly used techniques for diagnosing AK; however, they vary in accuracy, sensitivity, and speed. There does not appear to be one single best tool for AK diagnosis. Rather, a combination of diagnostic assessments should be performed to verify the presence of AK [28].

### 3.1. Clinical Examination

There are a number of clinical manifestations associated with AK. In early stages of the disease, signs include a grey-dirty epithelium, pseudodendritiformic epitheliopathy, multifocal stromal infiltrates, and, later, perineuritis, a ring infiltrate (Figure 1) [27,29]. Progressive diseases can include scleritis, iris atrophy, anterior synechiae, secondary glaucoma, mature cataract, and chorioretinitis [29]. Keratoneuritis or radial nerve enlargement may also occur [30,31]. Corneal endotheliitis has been reported as an uncommon manifestation of AK [32].

Clinical manifestations of AK may resemble other types of infectious keratitis (i.e., fungal, bacterial, or herpes simplex), making diagnosis purely on the basis of clinical examination difficult [22,32]. To confirm the diagnosis, other diagnostic tests such as PCR, culture, or in vivo confocal microscopy (IVCM) are employed. Additionally, coinfection with other microorganisms can occur, making treatment more difficult [33,34].

### 3.2. Polymerase Chain Reaction

Polymerase chain reaction (PCR) is a molecular technique used to rapidly amplify DNA samples. In the context of AK, it is used as a means of diagnosis by detecting the genetic material (DNA) of *Acanthamoeba* in ocular surface samples [35]. PCR is highly sensitive and specific, enabling the detection of even low concentrations of *Acanthamoeba* DNA. Its rapid result aids in early diagnosis and timely treatment initiation to prevent the infection from progressing.

There have been several PCR assays used to target *Acanthamoeba*. More specifically, they all target different regions of the nuclear small subunit 18S rRNA gene [36,37,38,39,40]. Of the papers reviewed in this study, nine mentioned the method and primers that were used for PCR amplification. The primers established by Schroeder et al. were the most commonly used, which amplify the 18s rRNA gene by making use of the JDP1 and JDP2 primers: (5′-GGCCCAGATCGTTTACCGTGAA-3′) and (5′-TCTCACAAGCTGCTAGGGAGTCA-3′) [14,36,40,41,42]. Khosravinia et al. compared the JDP primers established by Schroeder et al. with the Nelson primers established by Mathers et al. as well as culture and smear, and found that the JDP primers yielded the most accurate and sensitive method of AK diagnosis [43]. Others have sought to develop new PCR assays with which to diagnose AK [38,44,45]. For instance, Lamien-Meda et al. developed a real-time PCR assay to identify only the T4 genotype, the most commonly identified genotype of *Acanthamoeba*, to provide greater specificity and speed in diagnosis [46]. There does not appear to be any one single PCR assay which performs better than the rest. However, a combination of PCR assays can be run in a parallel in an effort to improve diagnostic sensitivity [47].

One study sought to examine the efficacy of next-generation sequencing (NGS) for high-throughput DNA sequencing [48]. Relative to real-time PCR, the specificity and sensitivity of the NGS assay were 100% and 88%, respectively [49]. While it performs only slightly worse than PCR and costs more, its ability to detect genetic variants of species as well as detect coinfections may point to its use as a potential diagnostic method in the future.

PCR is a valuable technique for diagnosing AK, but it does have limitations. Its sensitivity and specificity, although generally high, can be influenced by factors like sample quality and the genetic similarity of *Acanthamoeba* to other microorganisms [48]. The variability in *Acanthamoeba* DNA concentration in clinical samples can lead to false negatives, and obtaining adequate corneal samples for testing can be challenging [35]. Additionally, laboratory expertise is required, and the method might not differentiate between active infection and non-viable organisms. The cost, time, and the potential inability of some PCR assays to detect emerging strains further contribute to limitations. Therefore, PCR is often complemented with other diagnostic methods to enhance accuracy in AK diagnosis.

### 3.3. Culture and Staining

Stains and cultures have long been considered the “gold standard” for the diagnosis of AK and other types of microbial keratitis [27,28,48]. By isolating and growing live *Acanthamoeba*, culture can provide unequivocal evidence of active infection. Additionally, culture enables the identification of specific *Acanthamoeba* species or strains, contributing to a better understanding of the disease’s characteristics and potential virulence [8]. For instance, *Acanthamoeba castellanii* and *Acanthamoeba polyphaga* are two of the most common species which cause AK [50]. This information is important in guiding appropriate treatment strategies as various AK species may react differently to certain medications or treatment methods [51]. Moreover, cultured *Acanthamoeba* can undergo antimicrobial susceptibility testing, aiding in the selection of effective therapeutic agents. It can also be accomplished faster than some PCR tests [52].

Bacteria are a primary energy source for *Acanthamoeba* trophozoites, so cultures are often grown on non-nutrient agar plates covered with *E. coli* [28,50]. Blood agars including sheep, horse, or chocolate are also used, and Sabourad agar is utilized when there is clinical suspicion for fungal keratitis. Because the initial clinical picture for AK is often non-specific, all of these cultures are often taken. Calcofluour white (CFW) and potassium hydroxide staining are commonly used for smears when there is suspicion of *Acanthamoeba*, though others have also been utilized, including hematoxylin and eosin (H and E), periodic acid-Schiff (PAS), and Gomori methanamine silver (GMS) [48]. A combination of staining media may yield more true positive results.

Two studies in this review made use of CFW staining. A study of 43 patients found that a positive CFW test can be used as a definitive confirmation of AK, given virtually no false positives were observed. However, in the case of a negative CFW test, there is not enough information to rule out the possibility of AK. The second study made use of CFW in combination with an agar culture for the confirmation of AK in a patient with HIV [53]. KOH mount is often used in place of CFW in resource-poor settings or where a fluorescence microscope is unavailable. The main disadvantage of KOH is its transparent nature which can make the visibility and identification of organisms, including *Acanthamoeba*, difficult. Dyes can be added to the preparation to aid in visualization. Smears are used in combination with cultures; smears are significantly less sensitive than cultures but have a rapid turnaround time and can direct early treatment in the setting of a positive result [54].

Culture has its limitations, including its time-consuming nature, potential for false negatives due to low sensitivity, and the need for specialized expertise and equipment [48]. A broad range of sensitivity for cultures between 33 and 67% has been reported [55]. Final culture results can also take upwards of 1 week in a disease where any delay in treatment can result in worse visual outcomes. The method of sample collection can also affect sensitivity, with a recent study demonstrating the highest probability of positive culture from a modified bent 21 g needle, followed by collection with a scalpel or conventional needle, and then, lastly, a cotton swab [56]. In cases of negative cultures and continued suspicion of AK, or for deeper disease (i.e., intrastromal), a corneal biopsy can be performed for microbiologic and histopathologic evaluation, and has been shown to yield positive results in the setting of negative epithelial cultures [57]. Despite its limitations, culture remains an essential component of the diagnostic toolkit for AK, and in practice, a combination of diagnostic methods is often employed to ensure accurate diagnosis, with methods such as PCR, immunofluorescence assays, and IVCM increasingly used for their more rapid results and higher sensitivities, as described in this review.

### 3.4. IVCM

In vivo confocal microscopy (IVCM) is a non-invasive imaging technique used to diagnose and monitor AK. By capturing high-resolution images of the cornea’s layers, IVCM enables the direct visualization of structures, aids in detecting *Acanthamoeba* trophozoites and cysts, and assesses infection severity. IVCM’s real-time imaging helps differentiate AK from similar conditions, track treatment progress, and provide longitudinal data. While its availability varies, IVCM, along with other diagnostic methods, enhances accuracy in diagnosing AK and guiding treatment decisions.

The two main types of IVCM include laser devices such as Heidelberg Retina Tomograph III (HRT) and white light devices such as Confoscan (Nidek, Japan) [58]. While there are differences in the way these devices work, they both have been used for monitoring and diagnosing AK in patients [58,59,60]. The speed of confocal microscopy is particularly advantageous and can allow for early treatment and improved clinical outcomes [61].

AK can present itself in both the cystic and trophozoite forms. In its cystic form, *Acanthamoeba* appears as a highly reflective nucleus surrounded by a cell wall and dark rings (Figure 2) [59]. They can also take irregular shapes such as triangles, asterisks, or hollow rings. The cysts may also form chains of three–six units. Following antimicrobial drug treatment, the cell walls of the cysts begin to dissolve, and a black hole presents itself around some of the dissolved cysts.

One study which compared the outcomes of cases of AK with and without the use of IVCM found that there was a significant delay if only culture was used, and that IVCM can prove useful in accelerating AK diagnosis [62]. Using IVCM as an adjunct tool for AK diagnosis in addition to culture led to significantly better patient outcomes compared to cultures alone. Another study found that in comparison to PCR and culture, IVCM was the most accurate tool for AK diagnosis [28].

Another study presented three cases in which IVCM led to improved patient outcomes and more informed clinical decision making [63]. Specifically, amongst these three cases, IVCM served as a secondary diagnostic when initial PCR results were inconclusive, aided in adjusting initial treatment until the complete resolution of AK, confirmed residual fungal infections, and helped clinicians understand early immune changes in the sub-basal corneal nerve plexus.

Although IVCM has proven useful as an adjunct tool in the diagnosis of AK, there exists subjectivity in the grading of confocal microscopy images [64]. Thus, the efficacy of IVCM can vary greatly depending on the operator as well the grader. Furthermore, the corneal region and area which IVCM captures is limited, and this may result in scans being taken in regions unaffected by AK. In cases of stromal inflammation, false negatives may arise due to the masking of *Acanthamoeba* cysts by inflammatory cells and edema. Conversely, false positives can occur if macrophages are incorrectly identified as *Acanthamoeba* cysts.

IVCM can be an invaluable tool for diagnosing and monitoring the progression of AK, but its use should be considered in light of the available resources, expertise, and clinical requirements, and should be employed in combination with other diagnostics such as PCR and culture [28].

### 3.5. Antibody

Antibody-based tests for AK diagnosis are an emerging approach that can provide valuable information about the presence of the pathogen and the immune response of the patient. Antibody-based tests are often used to detect specific structures of *Acanthamoeba* spp. that are indicative of AK or recognize antibodies produced by the immune system in response to an *Acanthamoeba* infection.

Several studies have reported on target structures in *Acanthamoeba* spp. These potential targets for antibodies may aid in diagnosis in the future: inosine-uridine preferring nucleoside hydrolase (IPNH), chorismite mutase (CM), carboxylesterase, adenylyl cyclase-associated protein (ACAP), and periplasmic binding protein (PBP) [65,66,67,68,69]. Additional studies have demonstrated the utility of CM and ACAP/PBP antibodies in AK diagnosis in mouse and rabbit animal models, respectively [69,70]. Antibody-based tests hold potential in the diagnosis of AK but still require further investigation and development prior to their widespread use.

### 3.6. Machine-Learning-Based

Two studies included in the search criteria sought to develop new machine/deep learning-based approaches for AK diagnosis. The first study by Zhang et al. created a deep learning model, KeratitisNet, for diagnosing and classifying infectious keratitis on slit-lamp images [71]. A dataset of 4830 slit-lamp images was used to train the model, and an additional 200 were reserved for external validation. The final model achieved an accuracy of 77.08%. The accuracy of KeratitisNet for diagnosing AK was 83.81%, and the Area Under the Receiver Operating Characteristic curve (AUROC) was 0.96. The model was compared to the observations of three corneal specialty ophthalmologists and demonstrated a significantly higher performance in diagnostic ability. Two additional studies, which were not included in the search criteria of the review but remain relevant, sought to accomplish similar endeavors. Xu et al. sought to use deep sequential feature learning for the image classification of infectious keratitis, but did not explicitly classify *Acanthamoeba* as its own group, instead aggregating it with other corneal diseases [72]. And, Koyoma et al. created an algorithm using a dataset of 4306 slit-lamp images with an accuracy/AUROC for *Acanthamoeba* of 97.9%/0.995 [73]. However, only a total of 19 cases of AK were included, possibly limiting the robustness of the model for AK diagnosis.

The second study that met the search criteria of this review sought to create a deep-learning framework for the diagnosis of fungal keratitis (FK) and AK using IVCM [74]. This study introduced the IVCM-Keratitis dataset, a dataset of 4001 IVCM images taken from the Central Eye Bank of Iran. The dataset was split with a 3000:1001 train/test split. The final model had a sensitivity of 91.4% and 97% for AK and FK, respectively, and a specificity of 98.3% and 96.4% for AK and FK, respectively. Given the specialized training required to evaluate IVCM images, as well as the tedious nature of assessing many IVCM images, such a model could serve as an adjunct tool to improve the speed and accuracy of diagnosis [75,76].

## 4. Treatment

There does not appear to be a single drug which can definitively remove both the cystic and trophozoite forms of *Acanthamoeba* [8,55]. While the trophozoite form is much more easily eliminated, the cystic form is often difficult to target and eliminate [77,78,79]. Early recognition and prompt diagnosis can be critical in AK treatment, preventing deeper layers of the cornea from being affected [12,80,81,82]. Treatment for AK typically consists of antimicrobial agents, including biguanides, diamidines, antiseptics, antiparasitics, antibiotics, and antifungal agents. In resistant disease, procedures such as amniotic membrane transplantation, photodynamic therapy, or keratoplasty may be employed [27]. Below, we outline the main treatments for AK and highlight potential treatments that may prove beneficial in combatting AK in the future.

### 4.1. Antimicrobial Agents

A combination of biguanides polyhexamethylene biguanide (PHMB) and chlorhexidine are the most common first-line treatment of AK [21,33,83,84]. Often they are used in combination but may also be used separately. Chlorhexidine and PHMB are extremely potent against *Acanthamoeba*; however, refractory cases do occur. These agents are also nonspecific and have cytotoxicity against corneal epithelial cells and keratocytes at clinically relevant doses. The most commonly used concentration of both PHMB and chlorhexidine is 0.02%. Clinically relevant doses may range from 0.02 to 0.06% for PHMB and 0.02 to 0.2% for chlorhexidine, with the use of higher concentrations being reported in refractory cases [4]. Until recently, the relative efficacy of various concentrations and combination therapy was unknown. In 2023, Dart et al. published a randomized controlled trial which demonstrated PHMB 0.08% monotherapy to be effective in treating AK, with a cure rate of >86%, and with a similar efficacy as combination therapy with PHMB 0.02% and propamidine 0.1% [85]. While this cure rate is significantly higher than that reported in other studies, the authors acknowledge that the promising results may be attributable to trial treatment delivery protocol.

Other guanidino-containing compounds may exert similar anti-acanthamoebal effects to PHMB, such as polyhexamethylene guanidine (PHMG), polyaminopropyl biguanide (PAPB), and guazatine, though these have not been tested clinically [86]. Other antimicrobial agents including diamidines such as propamidine isethionate and hexamidine have also been used [87,88]. Given a limited access to propamidine isethionate, pentamidine isethionate shows promise as an alternative given that it has demonstrated inhibitory effects on *Acanthamoeba* trophozoites and cysts in vitro [89].

Following unsuccessful treatment with first-line agents, oral miltefosine, an anti-parasitic agent, has been employed as an adjunctive therapy with promising results [90,91,92]. The development of a miltefosine-eluting contact lens device may allow for sustained miltefosine release to treat AK [93]. Of note, oral miltefosine has been associated with severe inflammatory reactions and may require the concomitant, judicious use of topical and oral corticosteroids [92,94,95]. Further investigation is needed to better evaluate its efficacy.

Antifungal agents have also demonstrated cysticidal effects against *Acanthamoeba*. Amphotericin and natamycin have historically been used as adjunct therapy in AK, but more recently, azoles have been employed. Azoles are a class of antifungal agents that target sterol 14a demethylase (CYP51). Since *Acanthamoeba* CYP51 has 31–35% of the sequence identity of fungal CYP51, azoles may have potent amoebicidal and cysticidal properties [96]. These include imidazoles (clotrimazole, miconazole, and ketoconazole) and triazoles (itraconazole, fluconazole, and voriconazole). While there is no consensus on the most effective azole therapy for AK, the delivery of a combination of oral and topical preparations has been reported. Posaconazole and voriconazole appear to have the greatest corneal penetration in both topical and oral forms [96]. While topical preparations need to be specially compounded, oral forms are readily commercially available. Another azole demonstrating promise in the treatment of AK was found in one in vitro study, which found that isavuconazole was able to kill trophozoites within 24 h and prevented the transformation of inactive *Acanthamoeba* cysts into their active trophozoite forms [88]. While one case report has demonstrated its clinical effectiveness, there are no additional data evaluating its corneal penetration or clinical efficacy [97].

### 4.2. Corticosteroids

The use of corticosteroids in the treatment of AK is a subject of controversy [98]. While steroids can reduce inflammation and alleviate symptoms, they also carry risks, including potentially exacerbating the infection, masking symptoms, increasing the risk of penetrating keratoplasty, worsening visual outcome, and promoting the survival of the parasite [22,98,99,100,101]. Steroids can also lead to complications such as increased intraocular pressure and cataract formation [102]. The decision to use steroids in AK should be made with caution and be tailored to the specific clinical situation.

In one study of 194 patients, corticosteroid use prior to anti-amoebic therapy was found to be a risk factor for developing severe inflammatory complications. Another study of 224 patients displayed a positive correlation between corticosteroid use and treatment failure [103]. When steroid use occurs as a result of a misdiagnosis of AK, disease severity can be severely exacerbated [101,104]. While topical and oral steroids do have a role in reducing the disease burden of AK by reducing the associated inflammation, they must be used with caution and in the setting of definitive diagnosis and optimized anti-amoebal therapy.

### 4.3. Therapy Resistant Cases

Following unsuccessful outcomes with anti-amoebic therapy, other techniques have been employed to manage AK. These include keratoplasty, amniotic membrane transplantation, and cross-linking.

#### 4.3.1. Keratoplasty

In the case of AK, keratoplasty is considered when the infection has caused significant corneal damage and disease burden that cannot be effectively managed with medical treatments alone (Figure 3). It is reserved for advanced cases where other treatment options have been unsuccessful [18,105,106,107,108]. One study found that penetrating keratoplasty (PKP) within 5.3 months following the first symptoms of therapy-resistant AK can result in improved BVCA compared to delayed surgery [106].

On the other hand, a study comparing the clinical outcomes between therapeutic penetrating keratoplasty (TPK), therapeutic deep anterior lamellar keratoplasty (TDALK), and optical penetrating keratoplasty (OPK) on a total of 359 AK eyes found that OPK had the best clinical outcome. Of note, OPK was performed in an uninflamed eye for visual rehabilitation, once the infection had been managed by medical therapy [109]. This has been corroborated by other recent studies as well.108 In early stages of the disease, TDALK may be attempted, while later or more progressive stages require a PKP. Elliptical deep anterior lamellar keratoplasty has also been performed to focally reduce disease burden in early stages [110].

It is important to note the unique post-operative challenges that can occur after keratoplasty for AK [111]. One study of 59 eyes found that risk factors for recurrence include corticosteroid use prior to anti-amoebic therapy as well as the presence of hypopyon [111]. Common postoperative complications include graft failure, cataract, and uncontrolled glaucoma [107].

Another study involved the implementation of a new method which included performing a large diameter tectonic lamellar keratoplasty (TLK) followed by a central optical penetrating keratoplasty within the lamellar graft to treat cases of corneal opacity with vascularization and peripheral thinning since traditional keratoplasty may not be as advantageous for these patients [112].

#### 4.3.2. Amniotic Membrane Transplantation

Amniotic membrane transplantation (AMT) is an adjunctive treatment approach for AK, during which a piece of amniotic membrane is placed on the infected cornea to aid in healing, alleviate discomfort, and reduce inflammation [113,114]. The amniotic membrane’s bioactive components support tissue regeneration and help protect the cornea from further damage and scarring [115]. In one review examining the use of AMT for infectious keratitis, AMT was found to provide benefits in AK management in several case series [116].

#### 4.3.3. Photodynamic Therapy/Cross-Linking

Photo-activated chromophore for keratitis corneal cross-linking (PACK-CXL) has shown benefits as an adjuvant therapy for the treatment of AK [117,118,119]. PACK-CXL with both riboflavin and rose bengal has been found to exert significant anti-amoebic activity [117]. High fluence PACK-CXL treatment may also target disease burden in AK, leading to pain relief, re-epithelization, and elimination of *Acanthamoeba* [118]. In another study, PACK-CXL halted corneal melting in the setting of AK [120]. While further investigation is needed, PACK-CXL may be a useful adjunctive therapy in recalcitrant AK.

### 4.4. Machine-Learning

While many diagnostic and prognostic and microbiologic modalities utilizing machine-learning are under investigation, including by our group, there is only one published report in this regard. Conventional cysticidal assays to determine the effectiveness of various anti-microbial treatments against *Acanthamoeba* species involve treating cysts and then observing the culture manually to detect signs of excystation. However, this approach is slow, requires significant labor, and has limited capacity for high-volume testing. As such, Shing et al. developed a machine-learning-based cysticidal assay to recognize *Acanthamoeba* trophozoites and cysts in microscopy images in an effort to improve the speed at which new therapeutics can be discovered [121]. Our group is currently identifying ways in which machine-learning can be employed to automate the diagnosis of AK from confocal microscopy images with encouraging preliminary results. Several other groups are examining similar constructs. The successful implementation of machine-learning modalities has the potential to significantly reduce the time until diagnosis and treatment, improve the accuracy of diagnosis, and ultimately, improve patient outcomes.

## 5. Discussion

AK can have devastating ocular consequences and is often difficult to treat. Because diagnosis can be difficult in its early stages, and diagnostic modalities that do exist can take significant time to return a result, AK may go left untreated or suboptimally treated until diagnosis. Early diagnosis is a key measure in improving visual outcomes. Additionally, any one current treatment strategy cannot fully address the burden of AK. The emergence of novel diagnostic techniques and treatment modalities has the potential to significantly impact AK diagnosis and management. In this review, we explored recent advancements, focusing on publications in the last 5 years.

The introduction of molecular diagnostic methods, such as the polymerase chain reaction (PCR), has enhanced AK diagnosis with improved sensitivity and specificity. PCR-based techniques allow for the rapid identification of *Acanthamoeba* DNA, enabling the timely initiation of appropriate treatment regimens. In recent years, newer PCR techniques and primer combinations have allowed for easier and more sensitive diagnosis of AK. Moreover, the integration of advanced imaging technologies, such as IVCM, has facilitated the precise visualization of corneal structures and *Acanthamoeba* cysts, aiding clinicians in disease diagnosis and staging and in monitoring treatment response. Employing machine-learning algorithms for the automation of diagnosis through IVCM holds great potential for improving AK diagnosis.

Recent advances in therapeutic options for AK also have the potential to improve outcomes in AK with more widespread and evidence-based use. Standard topical therapy for AK has remained stagnant for decades until very recently where the employment of newer agents with anti-amoebal properties has shown promise. Oral miltefosine and topical and oral azole anti-fungals have demonstrated clinical potential against both trophozoite and cyst stages, though further study is needed. Additionally, the development of topical formulations and controlled-release drug delivery systems that can adequately penetrate the cornea holds the prospect of enhancing treatment outcomes through greater drug bioavailability, sustained therapeutic levels, and reduced dosing frequency.

Lastly, the advent of PACK-CXL as an adjuvant therapy for AK represents a potential therapeutic strategy with an improvement in treatment protocols.

Long-term studies assessing the efficacy and safety of emerging treatments are necessary to establish evidence-based therapeutic guidelines. Moreover, the economic feasibility and accessibility of these newer diagnostic methods and treatments must be considered to ensure their widespread applicability.

## 6. Conclusions

The recent progress in diagnosing and treating AK holds great promise in improving patient outcomes and reducing the burden of this sight-threatening infection. Collaborative efforts between researchers, clinicians, and pharmaceutical industries are vital to further refine these innovations and address the unmet needs in this field. Future research should focus on optimizing diagnostic and treatment modalities, exploring and implementing new diagnosing modalities, such as through machine-learning, and expanding access to advanced diagnostic and therapeutic tools to comprehensively combat *Acanthamoeba* keratitis.

## Figures and Tables

**Figure 1 biology-12-01489-f001:**
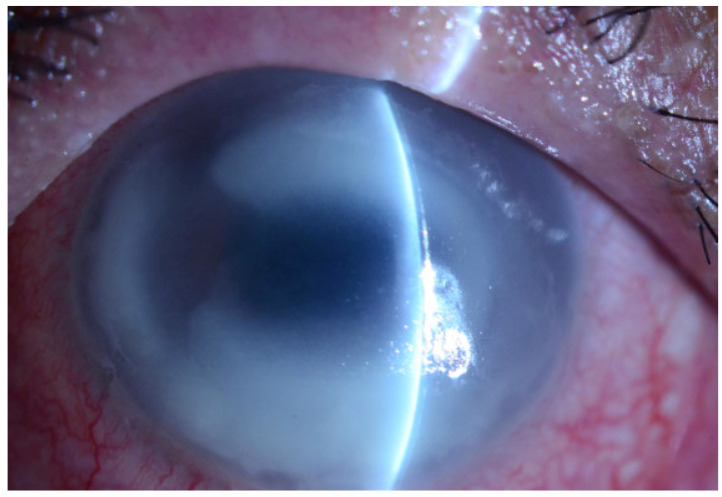
Slit lamp photo of ring infiltrate of the left cornea.

**Figure 2 biology-12-01489-f002:**
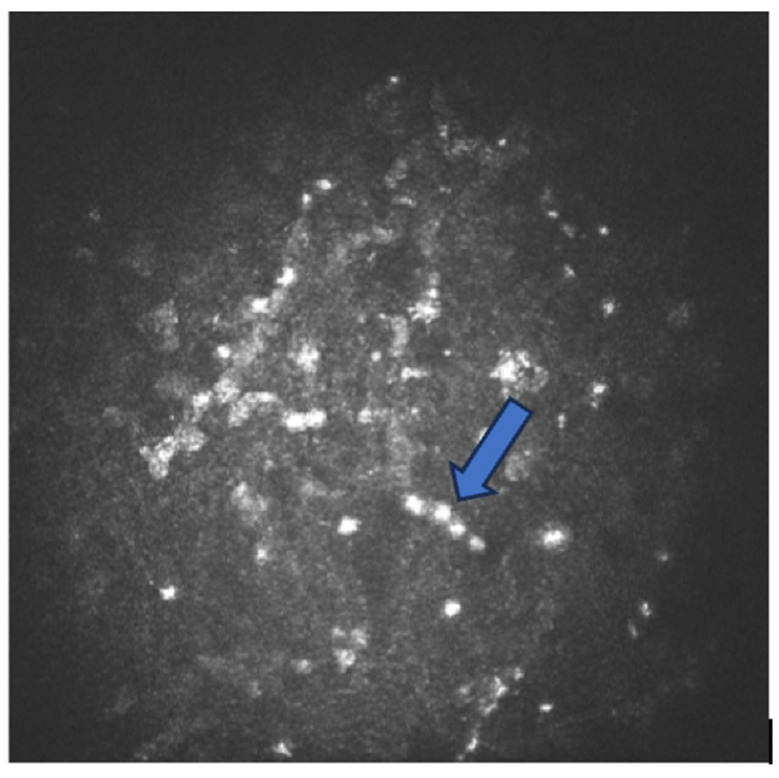
Heidelberg Retina Tomograph III (HRT) confocal microscopy image of amoeba cysts in a chain (blue arrow). The cysts have a highly reflective nucleus surrounded by a low-refractile ring.

**Figure 3 biology-12-01489-f003:**
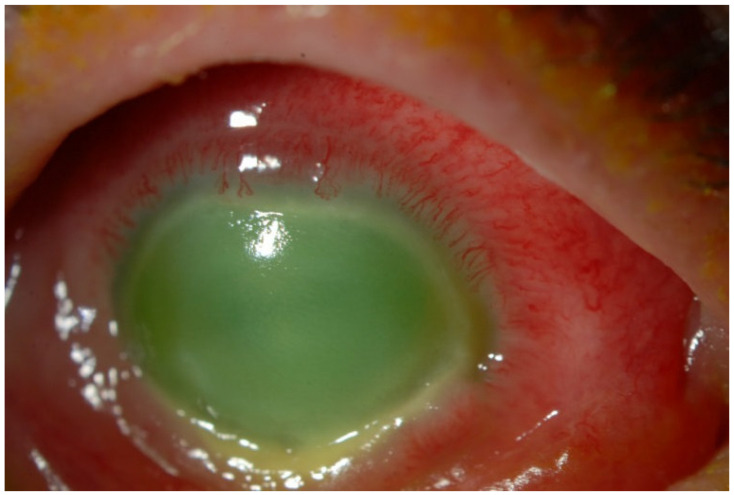
Slit lamp photograph showing an eye with severe *Acanthamoeba* keratitis and associated inflammation, unresponsive to medical therapy, and requiring therapeutic penetrating keratoplasty.

## Data Availability

No new data was generated in this study, and all data presented can be found by conducting a search of the cited articles on PubMed.

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
