# Peer review of "New Frontiers in Acanthamoeba Keratitis Diagnosis and Management"

_biology, 2023, doi:10.3390/biology12121489_

Round 1
Reviewer 1 Report
Comments and Suggestions for Authors
The paper provides a comprehensive overview of recent advancements in the diagnosis and treatment of Acanthamoeba keratitis (AK). The authors set the stage for understanding the significance of AK as a severe eye infection and the challenges associated with its diagnosis and treatment, outlining a systematic approach to literature review.The discussion of different diagnostic methods is insightful and presents a balanced view of the strengths and limitations of each method. The section on treatment options provides a detailed discussion of antimicrobial agents, corticosteroids, and treatment-resistant cases, including keratoplasty, amniotic membrane transplantation, and photodynamic therapy/cross-linking.
Specific Comments:
In the Methods section, it would be helpful to further specify the inclusion and exclusion criteria for selecting the literature. Additionally, it might be worth the final number of publications included in the review.
In the Treatment section, it would be valuable to include the most recent published results from the ODAK trial about the monotherapy with PHMB 0.08%.
Dart JKG, Papa V, Rama P, Knutsson KA, Ahmad S, Hau S, Sanchez S, Franch A, Birattari F, Leon P, Fasolo A, Kominek EM, Jadczyk-Sorek K, Carley F, Hossain P, Minassian DC. The Orphan Drug for Acanthamoeba Keratitis (ODAK) trial: PHMB (polihexanide) 0.08% and placebo versus PHMB 0.02% and propamidine 0.1. Ophthalmology. 2023 Oct 4:S0161-6420(23)00710-8. doi: 10.1016/j.ophtha.2023.09.031. Epub ahead of print. PMID: 37802392.
Overall, this paper is informative and well-structured. Addressing the specific comments above will further enhance its quality and readability.
Author Response
In the Methods section, it would be helpful to further specify the inclusion and exclusion criteria for selecting the literature. Additionally, it might be worth the final number of publications included in the review.
In the Treatment section, it would be valuable to include the most recent published results from the ODAK trial about the monotherapy with PHMB 0.08%.
Dart JKG, Papa V, Rama P, Knutsson KA, Ahmad S, Hau S, Sanchez S, Franch A, Birattari F, Leon P, Fasolo A, Kominek EM, Jadczyk-Sorek K, Carley F, Hossain P, Minassian DC. The Orphan Drug for Acanthamoeba Keratitis (ODAK) trial: PHMB (polihexanide) 0.08% and placebo versus PHMB 0.02% and propamidine 0.1. Ophthalmology. 2023 Oct 4:S0161-6420(23)00710-8. doi: 10.1016/j.ophtha.2023.09.031. Epub ahead of print. PMID: 37802392.
Overall, this paper is informative and well-structured. Addressing the specific comments above will further enhance its quality and readability.
Thank you for this positive review. We have added to the Methods section in lines 96-102.
Thank you for mentioning the ODAK trial which significantly enhances recent advances in the treatment of AK. This will be the first review to comment on this study. We have added this information to lines 318-323.
Reviewer 2 Report
Comments and Suggestions for Authors
This is a review of the identification and treatment update on Acanthamoeba Keratitis. It is very well written and complete. A very few issues need to be addressed
In general, should the “in vivo” and “in vitro” phrases be italicized? What is the journal policy on this?
More information is needed in the figure legend of figure 2. A scale bar would be useful and the cysts should be pointed out using arrows as it’s not clear what we are looking at here?
In a number of the cited references, the author list has been truncated. This may be a data base artefact. A few examples are:-
Ref 5. The full reference is
Bacon, A. S., Frazer, D. G., Dart, J. K. G., Matheson, M. M., Ficker, L. A., & Wright, P. (1993). A review of 72 consecutive cases of Acanthamoeba keratitis, 1984–1992. Eye, 7(6), 719-725.
Ref 43. KHOSRAVINIA in capitals? And the full reference is
Khosravinia, N., Abdolmajid, F. A. T. A., Moghaddas, E., Farash, B. R. H., Sedaghat, M. R., Eslampour, A. R., & Jarahi, L. (2021). Diagnosis of Acanthamoeba keratitis in Mashhad, northeastern Iran: a gene-based PCR assay. Iranian Journal of Parasitology, 16(1), 111.
Ref 53. The full reference is
Li, S., Bian, J., Wang, Y., Wang, S., Wang, X., & Shi, W. (2020). Clinical features and serial changes of Acanthamoeba keratitis: an in vivo confocal microscopy study. Eye, 34(2), 327-334.
Ref 73. The full reference is
Lorenzo-Morales, J., Martín-Navarro, C. M., López-Arencibia, A., Arnalich-Montiel, F., Piñero, J. E., & Valladares, B. (2013). Acanthamoeba keratitis: an emerging disease gathering importance worldwide?. Trends in parasitology, 29(4), 181-187.
Ref 91. The full reference is
Nakagawa, H., Koike, N., Ehara, T., Hattori, T., Narimatsu, A., Kumakura, S., & Goto, H. (2019). Corticosteroid eye drop instillation aggravates the development of Acanthamoeba keratitis in rabbit corneas inoculated with Acanthamoeba and bacteria. Scientific Reports, 9(1), 12821.
Ref 103. The full reference is
Wang, H., Jhanji, V., Ye, C., Ren, Y., Zheng, Q., Li, J., Zhao, Z. and Chen, W., 2023. Elliptical deep anterior lamellar keratoplasty in severe Acanthamoeba keratitis. Indian Journal of Ophthalmology, 71(3), p.999.
Author Response
In general, should the “in vivo” and “in vitro” phrases be italicized? What is the journal policy on this?
Thank you. All instances of in vivo and in vitro have been italicized.
More information is needed in the figure legend of figure 2. A scale bar would be useful and the cysts should be pointed out using arrows as it’s not clear what we are looking at here?
Thank you. We have included a new figure demonstrating more classic cysts in a chain which are now indicated by an arrow.
In a number of the cited references, the author list has been truncated. This may be a data base artefact. A few examples are:-
Ref 5. The full reference is
Bacon, A. S., Frazer, D. G., Dart, J. K. G., Matheson, M. M., Ficker, L. A., & Wright, P. (1993). A review of 72 consecutive cases of Acanthamoeba keratitis, 1984–1992. Eye, 7(6), 719-725.
Ref 43. KHOSRAVINIA in capitals? And the full reference is
Khosravinia, N., Abdolmajid, F. A. T. A., Moghaddas, E., Farash, B. R. H., Sedaghat, M. R., Eslampour, A. R., & Jarahi, L. (2021). Diagnosis of Acanthamoeba keratitis in Mashhad, northeastern Iran: a gene-based PCR assay. Iranian Journal of Parasitology, 16(1), 111.
Ref 53. The full reference is
Li, S., Bian, J., Wang, Y., Wang, S., Wang, X., & Shi, W. (2020). Clinical features and serial changes of Acanthamoeba keratitis: an in vivo confocal microscopy study. Eye, 34(2), 327-334.
Ref 73. The full reference is
Lorenzo-Morales, J., Martín-Navarro, C. M., López-Arencibia, A., Arnalich-Montiel, F., Piñero, J. E., & Valladares, B. (2013). Acanthamoeba keratitis: an emerging disease gathering importance worldwide?. Trends in parasitology, 29(4), 181-187.
Ref 91. The full reference is
Nakagawa, H., Koike, N., Ehara, T., Hattori, T., Narimatsu, A., Kumakura, S., & Goto, H. (2019). Corticosteroid eye drop instillation aggravates the development of Acanthamoeba keratitis in rabbit corneas inoculated with Acanthamoeba and bacteria. Scientific Reports, 9(1), 12821.
Ref 103. The full reference is
Wang, H., Jhanji, V., Ye, C., Ren, Y., Zheng, Q., Li, J., Zhao, Z. and Chen, W., 2023. Elliptical deep anterior lamellar keratoplasty in severe Acanthamoeba keratitis. Indian Journal of Ophthalmology, 71(3), p.999.
Thank you. Indeed, this was an artefactual error. All of these references have now been updated.
Reviewer 3 Report
Comments and Suggestions for Authors
The Authors should add some information about Acanthamoeba species involved in AK, allowing to better understand the statement at line 132 (ref8).
Moreover, further information should be given regarding culture media and techniques
lines 151 and 276- in vivo in italics
Author Response
The Authors should add some information about Acanthamoeba species involved in AK, allowing to better understand the statement at line 132 (ref8).
Thank you. This information has now been added to lines 168-172.
Moreover, further information should be given regarding culture media and techniques
We have significantly added to the culture and staining section. New information is included in lines 175-183, 188-194, 197, and 199-205.
lines 151 and 276- in vivo in italics
This has been changed.